# The State of Pharmacoeconomics Education in the Doctor of Pharmacy Curriculum amid the Changing Face of Pharmacy Practice

**DOI:** 10.3390/healthcare11222923

**Published:** 2023-11-08

**Authors:** Georges Adunlin, Jordan Skiera, Chandler S. Cupp, Askal Ayalew Ali, Serge Amani Yao Afeli

**Affiliations:** 1Department of Pharmaceutical, Social and Administrative Sciences, McWhorter School of Pharmacy, Samford University, Birmingham, AL 35229, USA; 2McWhorter School of Pharmacy, Samford University, Birmingham, AL 35229, USA; jskiera@samford.edu (J.S.); ccupp@samford.edu (C.S.C.); 3Economic, Social and Administrative Pharmacy (ESAP), College of Pharmacy and Pharmaceutical Sciences, Institute of Public Health, Florida A&M University, Tallahassee, FL 323107, USA; askal.ali@famu.edu; 4Department of Pharmaceutical and Administrative Sciences, Presbyterian College School of Pharmacy, Clinton, SC 29325, USA; safeli@presby.edu

**Keywords:** pharmacoeconomics, pharmacy education, teaching, health outcomes research, United States

## Abstract

(1) Background: Continuous growth in pharmaceutical expenditure indicates the need for more advanced pharmacoeconomics evaluations to optimize healthcare outcomes and resource allocation. This study assesses the extent to which accredited pharmacy colleges in the United States cover pharmacoeconomics content within the didactic curriculum of their Doctor of Pharmacy (PharmD) programs. (2) Methods: We conducted a systematic search of the websites of accredited professional-degree programs in pharmacy schools located in the United States to identify pertinent content related to pharmacoeconomics. (3) Results: Out of 141 pharmacy programs, a total of 111 programs of varying sizes were found to have publicly accessible information regarding the content of their pharmacoeconomic curricula on their websites. All these programs required the inclusion of pharmacoeconomics content in their curricula. An examination of course syllabi revealed that the goals and descriptions were broad, aiming to provide students with an introductory understanding of the principles of pharmacoeconomics. The number of credit hours allocated to pharmacoeconomics education ranged from one to seven across the programs. The approach to delivering pharmacoeconomics content varied among the programs. (4) Conclusions: Advanced knowledge of the principles of pharmacoeconomics must be an integral component of all PharmD curricula to prepare pharmacists to assess the rational use of pharmacy products and services, improve clinical outcomes, and mitigate healthcare expenditures.

## 1. Introduction

Pharmacoeconomics is a sub-discipline of health economics that focuses on the economic evaluation of pharmaceutical products, treatments, and healthcare interventions [1]. It involves the application of economic principles and methods to assess the value, efficiency, and outcomes associated with various healthcare interventions, particularly in the context of pharmaceuticals [2].

The major topics involved in pharmacoeconomics include cost-effectiveness, cost-minimization, cost-utility, and cost-benefit analyses. It was not until the emergence of “evidence-based” medicine that notable advances were achieved in the use of pharmacoeconomics to identify, measure, and compare the costs and consequences of pharmaceutical products and services. Advances in new technologies have led to an exponential increase in competing treatments for a wide range of diseases [3,4]. Consequently, healthcare use and expenditures have gradually increased. Due to changing demographics and rising healthcare costs, the United States government has emphasized the need for enhanced value evaluation in healthcare spending; hence, the United States has adopted various strategies to control rising healthcare expenditures [5,6], focusing on medication costs as a significant target for potential savings. Furthermore, pharmaceutical companies in the United States have been required to submit economic studies on the implications of introducing new drugs to the Food and Drug Administration (FDA) [7].

As value-based care becomes increasingly relevant in patient care, pharmacoeconomics continues to gain prominence within the clinical setting [8], yet the subject occupies an uncertain place within professional pharmacy programs [9,10,11]. Rascati and colleagues conducted a study examining the extent of pharmacoeconomics education provided during the 1996–1997 academic year [9]. Of the 79 schools surveyed, 63 (80%) provided pharmacoeconomics education at the BS and/or PharmD level. Subsequent surveys conducted in 2007 and 2013 found that, while pharmacoeconomics education is provided in nearly all United States pharmacy programs, there is variation in the topics covered and in teaching hours [10,11]. Given that these studies are dated, there is currently a lack of information regarding the content of pharmacoeconomics taught within the current PharmD curriculum in United States-based pharmacy programs. Meanwhile, accreditation bodies, educational councils, and professional pharmacy organizations have consistently promoted pharmacoeconomics education and knowledge within the curricula of United States-based pharmacy programs for several years. The 2016 Accreditation Council for Pharmacy Education (ACPE) outlined the required elements of the didactic PharmD curriculum, including the following description of pharmacoeconomics as the “application of economic principles and theories to the provision of cost-effective pharmacy products and services that optimize patient-care outcomes, particularly in situations where healthcare resources are limited“ [12]. Since 2010, the North American Pharmacist Licensure Examination has consistently included pharmacoeconomics competency statements [10,13]. Previously, the International Society for Pharmacoeconomics and Outcomes Research (ISPOR) Educators’ Toolkit Task Force provided valuable teaching resources for pharmacoeconomics educators. In 2020, ISPOR introduced the ‘New Competencies Framework for Health Economics and Outcomes Research Professionals’ [14], which serves as a guide for academic curricula, fellowships, and continuing-education programs and for the evaluation of candidates for health economics and outcomes research (HEOR) career opportunities.

The role of pharmacists has extended beyond medication distribution to promoting sound and cost-effective drug therapy, enhancing patient outcomes, and contributing to the overall improvement of healthcare in the community [15,16,17,18]. Specifically, with pharmacists’ knowledge of pharmacology, disease, anatomy, drug calculations, dosage, and drug–drug interactions, integrating pharmacoeconomics knowledge enables them to make informed therapeutic decisions. The acquisition of knowledge in pharmacoeconomics should allow pharmacists to assess and compare clinical guidelines, evaluate drug therapy recommendations, review data from clinical trials, and assess the economic impact of medications [16]. Hence, the potential role of pharmacists in integrating pharmacoeconomics into clinical practice is paramount [19]. As the healthcare industry experiences continuous innovation, the responsibilities of pharmacists and their role as pharmacoeconomics experts are also impacted, affecting the skillset required to adapt to the changing environment. Pharmacy programs must address the increasingly complex and specialized skills that employers desire. This study aims to assess the extent to which accredited pharmacy colleges in the United States cover pharmacoeconomics content within the didactic curricula of their Doctor of Pharmacy (PharmD) programs.

## 2. Materials and Methods

We systematically searched the curriculum websites of all 141 United States-based pharmacy programs holding accredited (full or candidate status) professional-degree programs to identify relevant pharmacoeconomics content. Information regarding pharmacoeconomics-related course descriptions, the number of credit hours, and syllabus content (if available) were determined for each pharmacy program and abstracted into a database. The data collection excluded graduate, post-doctoral, and fellowship-training programs, as the study focused exclusively on the PharmD degree. Our search did not cover elective courses or syllabi from experiential rotations because our assessment strictly focuses on required courses in the core curriculum. The location of the pharmacy programs was regionally classified based on four geographic regions established by the United States Census Bureau: the Northeast, Midwest, South, and West. Microsoft Excel for Microsoft 365 (Version 2310) was used to analyze the data [20].

Our study qualifies as a literature review and does not involve human subjects; thus, it does not require ethical or Institutional Review Board approval.

## 3. Results

Out of the 141 pharmacy programs included in the analysis, publicly available information on pharmacoeconomics content was identified on the websites of 111 programs. The relevant data from these programs were compiled in an Excel database, alphabetically listed for ease of reference. Of the included pharmacy programs, 57 (51%) were public institutions; 54 (49%) were private (Figure 1). Geographically, most of the assessed programs, 41 (37%), were in the South region. Notably, six of the pharmacy programs offered distance pathways or online PharmD education options.

While some information regarding the curriculum was publicly accessible, a smaller proportion of pharmacy programs made their syllabi available online. Out of the 111 schools assessed, only 30 syllabi were retrieved from online sources. It is noteworthy, however, that all the pharmacy programs included in our database required the inclusion of pharmacoeconomics content within their curricula. Figure 2 shows that pharmacoeconomics teaching predominantly occurred during the third professional year (*n* = 55; 51%) and the second professional year (*n* = 39; 36%). All the required pharmacoeconomics courses exhibited similarities in terms of their learning objectives. A comprehensive list of these learning objectives is outlined as follows:Define pharmacoeconomics;Identify and determine the relevant costs and consequences associated with pharmacy products and services;Define the differences between cost-benefit analysis (CBA), cost-effectiveness analysis (CEA), cost-minimization analysis (CMA), and cost-utility analysis (CUA);Discuss the significance of specifying/selecting perspectives for inclusion in the analysis;Overview the steps involved in conducting a pharmacoeconomic analysis;Critically evaluate current pharmacoeconomics literature;Identify the potential applications of pharmacoeconomics in various pharmacy settings.

Course descriptions across the analyzed syllabi were generally broad and primarily focused on equipping students with introductory concepts or the principles of pharmacoeconomics. Nonetheless, there is a comprehensive consensus among several course descriptions that balancing the cost with the consequences (outcomes) of pharmaceutical therapies and services is the overarching goal of pharmacoeconomics in PharmD education.

Figure 3 shows that the number of credit hours allocated to pharmacoeconomics education varied from one to seven. The majority of programs assigned either two credit hours (*n* = 54) or three credit hours (*n* = 45) to pharmacoeconomics topics; yet it is worth highlighting that those credit hours are not exclusive to pharmacoeconomics and are distributed among several topics integrated within one course. The amount and nature of the delivery of pharmacoeconomics or pharmacoeconomics-related content vary between programs. While most programs offered a required core course solely focused on pharmacoeconomics, others incorporated pharmacoeconomics alongside related subjects such as pharmacoepidemiology, population health, and pharmacy administration. Additionally, in some programs, pharmacoeconomics content was covered within a broader course encompassing various topics, including pharmacokinetics, pharmacodynamics, pharmacoepidemiology, and pharmacotherapy related to patient care.

The identified pharmacoeconomics topics were referenced within course descriptions, and the syllabi were categorized and presented based on 20 emerging topics, arranged in alphabetical order in Table 1. The majority of syllabi from the reviewed programs covered various methods of pharmacoeconomics analysis, although only about half (*n* = 54; 49%) of the reviewed programs’ syllabi mentioned the application of pharmacoeconomics. There were various ways to foster and assess students’ learning outcomes in the pharmacoeconomics course. Teaching methods varied and encompassed didactic lectures, guest speakers, videos, research projects, problem-based learning, case studies, and article-critiquing assignments. Notably, based on the information analyzed, no program that taught pharmacoeconomics in an interdisciplinary setting was identified. Reading assignments were prevalent in the reviewed syllabi, while several featured unique assignments designed to foster student engagement, ownership of learning, and productive interactions among peers. Such assignments included research projects, presentations, and group work. In the reviewed syllabi, skills were not identified as a significant part of the descriptions and objectives of the pharmacoeconomics courses.

As the sub-discipline of pharmacoeconomics has advanced, an extensive array of textbooks has become accessible. However, it is worth noting that many excellent pharmacoeconomics books authored outside the United States are inevitably shaped by the unique healthcare funding and provision systems of their respective countries. Consequently, their focus may not always align with the primary concerns and issues encountered in the United States healthcare system. Most pharmacoeconomics courses (*n* = 68; 61%) did not explicitly require a textbook. The courses that did require one were mostly assigned “Essentials of Pharmacoeconomics” by Karen Rascati [2]. Other courses assigned pharmacoeconomics chapter readings from other books related to pharmacy research, pharmacoepidemiology, and drug literature evaluation. Most of these readings are available through comprehensive online pharmacy resources, which several pharmacy schools subscribe to. Other readings consisted of reports, journal articles, and handouts, which were utilized to provide real-world pharmacoeconomics applications and to reinforce key concepts.

## 4. Discussion

A comprehensive literature search uncovered only three studies evaluating application-based or active learning in a pharmacoeconomics course within the United States PharmD curriculum [21,22,23]. Our evaluation of pharmacoeconomics curricula and instructional methods revealed a shift towards incorporating more active learning approaches; however, the impact of this shift has yet to be assessed. There are no resources or reports specifying the expectations of employers regarding pharmacoeconomics competencies in new PharmD graduates. Based on our professional assessment of the course syllabus, the skills provided were primarily foundational, especially considering the growing momentum surrounding pharmacoeconomics methods. Those foundational skills consist primarily of identifying the determinants of the health economy, different types of pharmacoeconomics evaluations, and critically analyzing pharmacoeconomics and outcome literature. To equip students with advanced-level skills, pharmacy programs should consider providing technical resources and training opportunities that emphasize practical application. Such advanced-level skills enable students to conduct trial- and modeling-based economic evaluations and indirect treatment comparisons, perform calculations involved in decision analysis, and engage in risk adjustment and Markov modeling.

The learning modality of pharmacoeconomics should encompass a combination of didactic training, laboratory training, and experiential learning [22,23,24]. Experiential learning is of particular importance as it is one of the most effective ways to enhance pharmacoeconomics education by engaging students through active learning strategies and fostering collaborations with industry exposure. Active learning sessions incorporate problem-solving exercises, case discussions, and laboratory activities. By actively participating in laboratory work, pharmacy students can develop a deeper understanding of the overall process involved in conducting pharmacoeconomics evaluation, as outlined by Jolicoeur and colleagues, which consists of ten steps: (1) defining the problem, (2) determining the study’s perspective, (3) determining the alternatives and outcomes, (4) selecting the appropriate pharmacoeconomics method, (5) placing monetary values on the outcomes, (6) identifying study resources, (7) establishing the probabilities of the outcomes, (8) applying decision analysis, (9) discounting costs or performing a sensitivity or incremental cost analysis, and (10) presenting the results, along with any limitations of the study [25]. Through these laboratory activities, students could learn how the results of pharmacoeconomics evaluations are translated and applied in clinical practice to support decisions in various areas, including individual patient treatment, disease management, drug-use guideline development, and formulary management. Additionally, pharmacy programs can significantly benefit from establishing educational partnerships with the industry. These partnerships can take the form of practical experiences or internships, providing students with valuable industry exposure, bridging the gap between classroom learning and real-world job experience, and equipping them with advanced skills and the knowledge necessary for successful careers in pharmacoeconomics.

Pharmacy education standards in the United States have undergone a substantial transformation, expanding beyond conventional basic science and clinical subjects. In light of recent developments in pharmacy education and the continuous implementation of curricular redesigns across these programs, [26,27,28], educators must formulate curricular strategies that are in harmony with the emerging trends in the field, which encompass aspects of pharmacoeconomics. The availability of data and information concerning pharmacoeconomics education in United States pharmacy programs is limited and needs to be updated. To accurately assess the current state of pharmacoeconomics education, our study examined the curriculum content of all accredited pharmacy programs in the United States, focusing on three key areas: course description, number of credit hours, and syllabus content (if available). Specifically, the assessment focused on core required courses rather than electives or experiential training, aiming to determine the extent to which all pharmacy students were exposed to pharmacoeconomics content during their PharmD program. The data we gathered reveal significant variations in the offerings of pharmacoeconomics programs within the PharmD curriculum, including variations in the number of credit hours, teaching hours, and the specific year in which the teaching occurred.

While pharmacoeconomics has been incorporated as a compulsory or elective course in schools or colleges of pharmacy in many countries, the variation observed in pharmacoeconomics education in the United States has also been noted globally [29,30]. A literature review examining the global trend of pharmacoeconomics courses in undergraduate pharmacy education revealed significant variability in course content [29]. Similarly, a study focusing on Eastern Mediterranean region pharmacy schools found a wide range of classroom hours dedicated to required courses covering pharmacoeconomics-related topics, varying from 2 to 60 hours among the different schools [30]. The fact that pharmacoeconomics instruction primarily occurs in the later stages of the didactic curriculum may provide students with the necessary problem-solving skills to apply pharmacoeconomics principles in clinical practice and guide clinical and policy decision-making. Since pharmacoeconomics goes beyond the economic evaluation of medical programs and pharmaceutical drugs, encompassing recommendations for efficiently utilizing healthcare resources, pharmacoeconomics knowledge must be reinforced through advanced pharmacy practice experiences or clinical rotations. Furthermore, because pharmacoeconomics takes a scientific approach to comparing the value of pharmaceutical products and services, its teaching has the potential to address various issues associated with drug affordability, such as drug access, medication adherence, patient outcomes, and health disparities.

Educational research within the pharmacy field has consistently demonstrated that evidence-based teaching and learning strategies positively impact student learning outcomes [31,32,33]. The fact that there is a lack of scholarship in teaching and learning (SoTL) focusing on pharmacoeconomics raises some questions about the priority given to this topic in pharmacy education. Engaging in SoTL focused on pharmacoeconomics can lead to evidence-based improvements in teaching practices, curriculum development, and student learning outcomes in this discipline. These efforts can also foster a culture of continuous improvement and innovation in pharmacoeconomics education in the PharmD curriculum. As the adoption of evidence-based instructional strategies becomes more widespread in pharmacy classrooms and practice, it becomes increasingly important to focus on the teaching and learning of pharmacoeconomics. Furthermore, there needs to be more sufficient data on PharmD students’ perceptions, motivations, and attitudes toward pharmacoeconomics in the United States. This highlights the need for pharmacy educators to make increased efforts to assess the impact of students’ behavior and their learning experiences related to pharmacoeconomics. By examining these factors, educators can gain valuable insights and further enhance the teaching and learning of pharmacoeconomics to meet pharmacy students’ needs effectively.

To the best of our knowledge, this is the only recent work that has investigated the state of pharmacoeconomics education in the United States regarding course objectives, number of credit hours, and syllabus content. However, our assessment does have some limitations worth acknowledging. First, several pharmacy programs did not publicly have their curriculum information posted on their main website, preventing us from assessing those programs. We did not contact the school representatives of such programs to obtain information regarding aspects of their curricula that were not accessible online. Second, our search methodology focused solely on identifying pharmacoeconomics-related content, meaning we may have missed other courses that cover concepts related to pharmacoeconomics but do not explicitly mention them. It is important to consider these limitations when interpreting the findings of our study. Future research could explore additional sources or methods to gather comprehensive data on pharmacoeconomics education in the United States, addressing these limitations and providing a more comprehensive understanding of the current state of pharmacoeconomics education. In future investigations, it would be crucial to longitudinally assess student learning and retention within the realm of pharmacoeconomics.

## 5. Proposed Recommendations

Taken individually, the results of this review lack the conclusive evidence needed to guide the comprehensive development or revision of a detailed plan for thoroughly integrating pharmacoeconomics into a pharmacy curriculum. Nonetheless, the results do point to essential principles that should be considered when educating pharmacy students to actively engage with pharmacoeconomics. They are as follows:The extensive implementation of problem-based learning in pharmacy schools offers a valuable opportunity for the seamless integration of pharmacoeconomics education across the curriculum. This integration will allow students to appreciate the subject’s significance in clinical practice and its fundamental role in shaping healthcare decision-making.Teaching needs to ensure that pharmacy students appreciate the relevance of learning the skills required to critically evaluate the effect of different healthcare interventions on patients. One way to accomplish this task is to introduce the concepts related to pharmacoeconomics early in the pharmacy curriculum and to build on and reinforce these concepts throughout the rest of the curriculum. The literature suggests that early one-off training is linked with poor knowledge in the long term [34,35]. When students are introduced to pharmacoeconomics and acquire related skills during their early years of PharmD education but do not continue to engage with this topic throughout the remainder of their training, they not only lose the acquired knowledge and skills but also any sense of their relevance.Reinforce pharmacoeconomics knowledge with relevant, real-world examples from peer-reviewed articles on topics of interest to students. For instance, learning about an incremental cost-effectiveness ratio could be embedded within consideration of a peer-reviewed article reporting the results of a clinical trial of combination antiretroviral therapy for human immunodeficiency virus.Patient cases spanning a diverse range of medical conditions across various practice settings could incorporate pharmacoeconomics principles. This integration challenges students to adopt a comprehensive perspective, considering not only the presented medical condition but also the pharmacoeconomics evidence underpinning treatment decisions.Illustrate how pharmacoeconomics and related knowledge/skills are used in daily pharmacy practice. Examples can be designed to demonstrate that pharmacoeconomics data can inform formulary decisions, aid in the selection or removal of drugs, and guide the development of practice guidelines aimed at promoting the cost-effective and appropriate utilization of pharmaceutical products. Other examples can explain that, in a community pharmacy setting, pharmacoeconomics can be applied by conducting a cost-effectiveness analysis of two common medications used to manage a chronic condition like hypertension. By comparing the total costs and health outcomes associated with Drug A and Drug B over a specified period, pharmacists can provide evidence-based recommendations to patients and healthcare providers, helping them make informed decisions about the most cost-effective treatment option while ensuring optimal health outcomes and resource utilization.Use of journal clubs in pharmacoeconomics education. Journal clubs are indispensable assets in pharmacoeconomics education for several fundamental reasons. To begin with, they serve as catalysts for critical thinking by necessitating students to dissect and appraise pharmacoeconomics research articles, thereby refining students’ capacity to scrutinize evidence and make well-informed decisions in clinical practice. Second, they act as a mechanism to ensure that students continuously stay abreast of the swiftly evolving realm of pharmacoeconomics, nurturing a culture of lifelong learning. Third, these clubs elevate communication proficiencies, empowering students to proficiently elucidate intricate pharmacoeconomic concepts to peers and fellow healthcare practitioners. Last, they foster collaboration and teamwork, which are indispensable competencies in healthcare environments that mandate interdisciplinary synergy for the provision of optimal patient care.Implement debates as a means to invigorate discussions on pharmacoeconomics subjects while acquiring proficiency in presentation techniques and critical assessment. A recent study demonstrated the efficacy of employing debates as an educational instrument, highlighting their capacity to augment students’ critical thinking abilities and aptitude for appraising pharmacoeconomics literature [36]. Moreover, the study reveals that this approach elevates students’ self-assurance in decision-making, enhances their critical analysis of evidence, and refines their communication skills for articulating arguments effectively.Students can gain advanced pharmacoeconomics knowledge and skills through comprehensive research projects offered within elective courses, capstone experiences, or summer research internships. These student-led research initiatives offer a unique opportunity for students to deepen their understanding of pharmacoeconomics, effectively connecting theoretical concepts to practical applications. These immersive experiences equip students with essential skills in data analysis, modeling, and the critical evaluation of healthcare interventions, making them well prepared for future challenges in the field.PharmD students aiming to acquire advanced pharmacoeconomics training beyond the standard didactic curriculum should have the opportunity to enroll in an advanced pharmacoeconomics elective. Typically, these electives are standalone courses that expand upon the foundational pharmacoeconomics curriculum. While introductory courses introduce students to fundamental pharmacoeconomics analysis methods, advanced pharmacoeconomics electives delve deeper, covering complex topics like multiple criteria decision analysis, discrete event simulations, Bayesian analysis, machine learning methods, artificial intelligence, value-based pricing, and health technology assessment. Through hands-on projects and case studies, students gain practical experience in conducting pharmacoeconomics evaluations, interpreting results, and effectively communicating their findings to stakeholders. These advanced courses equip future pharmacists with the research skills necessary to navigate the dynamic pharmaceutical industry and make substantial contributions to evidence-based practice.Tailor the instruction to a suitable level for students, acknowledging the wide-ranging educational backgrounds and experiences of PharmD students. Consider offering specialized tutoring sessions for students seeking extra support to ensure comprehensive support for all students.Pharmacoeconomics requires regular educational material updates for several crucial reasons. First, healthcare landscapes evolve rapidly, with new medications, technologies, and treatment approaches emerging frequently. Staying current ensures that students and professionals can apply the latest methodologies and data for accurate analysis and decision-making. Second, economic factors, such as inflation and changing healthcare policies, affect cost assessments and pricing strategies, necessitating adjustments in educational content. Moreover, updating materials fosters continuous learning and adaptability, enhancing the competency of practitioners in optimizing healthcare resource allocation and improving patient outcomes, ultimately reflecting the dynamic nature of pharmacoeconomics in modern healthcare.Establishing partnerships between academia and industry presents an opportunity to advance pharmacoeconomics knowledge within an academic context. Strategies like internships, industry mentors, on-site visits, and collaborative projects utilizing shared data are promising initiatives in this regard.

## 6. Conclusions

Training in pharmacoeconomics can be valuable for any healthcare professional who engages with patients and their families or communities. To understand the status of pharmacoeconomics education in professional pharmacy programs within the United States, we reviewed catalogs of 111 pharmacy programs. Despite the recognized importance of pharmacoeconomics in evidence-based care and drug-prescribing processes, our findings indicate uncertainty regarding the extent to which all pharmacy programs across the United States dedicate adequate credit hours to balance both the theoretical and practical considerations of pharmacoeconomics in their curricula. The pharmacy profession is quickly evolving, and today, pharmacists practice not only in traditional settings such as community pharmacies and hospitals but also in other settings requiring pharmacoeconomics knowledge. This requires pharmacy programs that plan to prepare today’s pharmacy students to meet tomorrow’s healthcare challenges by finding the appropriate balance between pharmacoeconomics course content, contact hours, and delivery methods that can meet the needs of students in realizing their future career goals.

## Figures and Tables

**Figure 1 healthcare-11-02923-f001:**
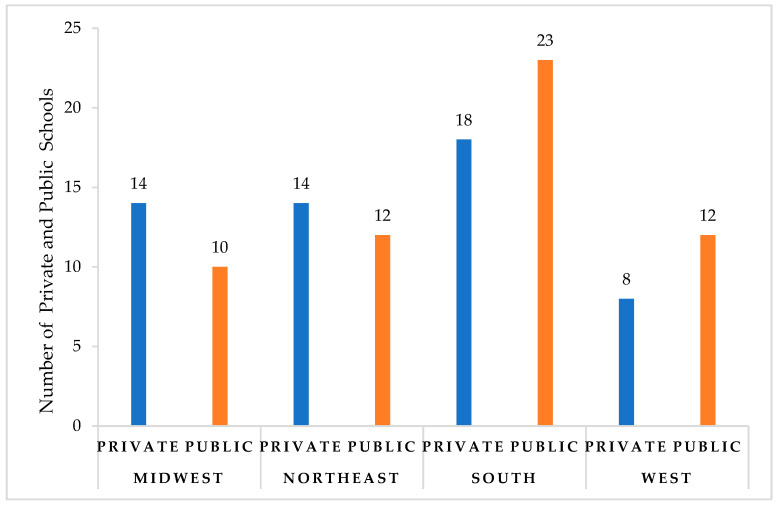
Distribution of assessed private and public pharmacy programs by region.

**Figure 2 healthcare-11-02923-f002:**
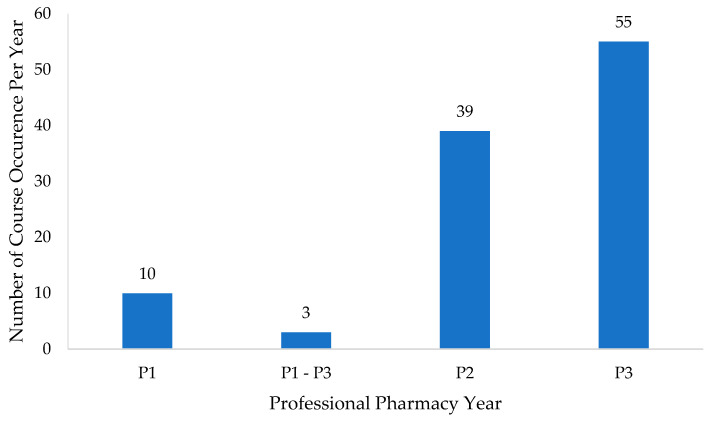
Pharmacoeconomics course occurrence by professional year (*n* = 107).

**Figure 3 healthcare-11-02923-f003:**
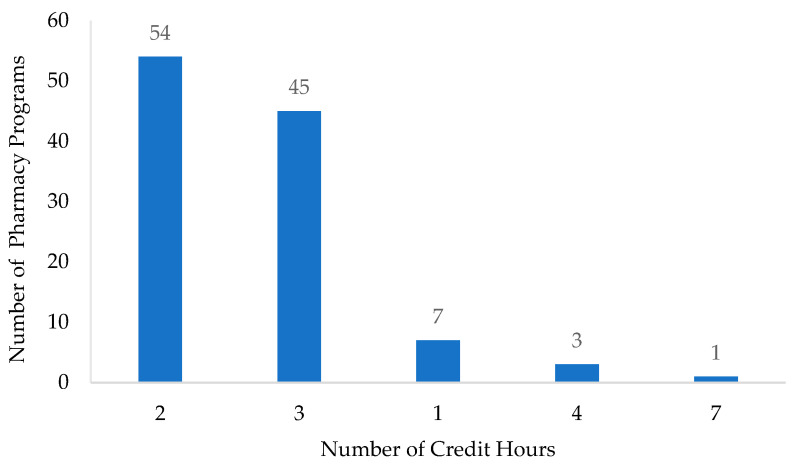
Semester credit hour distribution for pharmacoeconomics (*n* = 110).

**Table 1 healthcare-11-02923-t001:** Common topics covered in pharmacoeconomics courses.

1. Costs 2. Cost-benefit analysis (CBA) 3. Cost-effectiveness analysis (CEA) 4. Cost-minimization analysis (CMA) 5. Cost of illness (COI) 6. Cost-utility analysis (CUA) 7. Disability-adjusted life year (DALY) 8. Decision analysis 9. Effectiveness 10. Efficacy 11. Health-related quality of life (HRQOL) 12. Health technology assessment (HTA) 13. Markov model 14. Patient-reported outcomes (PROs) 15. Quality-adjusted life year (QALY) 16. Sensitivity analysis 17. Systematic review 18. Utility 19. Value-based pricing (VBP) 20. Willingness to pay (WTP)

## Data Availability

The data presented in this study are available upon request from the corresponding author.

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
