# Peer review of "The State of Pharmacoeconomics Education in the Doctor of Pharmacy Curriculum amid the Changing Face of Pharmacy Practice"

_healthcare, 2023, doi:10.3390/healthcare11222923_

Round 1

Reviewer 1 Report

Comments and Suggestions for Authors

This is an interesting review on Pharmacoeconomics education in the PharmD curriculum. It is comprehensive and clear for an international and diverse audience. 

Discussion part is a bit long, however an interesting list of suggestions is included. 

Limitations of the study are appropriately addressed. 

Page 7, line 253: avoid repetition of pharmacy programs

Author Response

This is an interesting review on Pharmacoeconomics education in the PharmD curriculum. It is comprehensive and clear for an international and diverse audience. Discussion part is a bit long, however an interesting list of suggestions is included. Limitations of the study are appropriately addressed. Page 7, line 253: avoid repetition of pharmacy programs.

AUTHORS RESPONSE:

Page 7, line 253: avoid repetition of pharmacy programs. The sentence was rewritten to address the reviewer comment.

“Given the recent changes in pharmacy education and the ongoing curricular redesigns implemented across such programs [26-28], they need to……”

Reviewer 2 Report

Comments and Suggestions for Authors

This is a good study evaluating the extent and topics of the pharmacoeconomics courses in PharmD programs in the US. A few minor comments are suggested to be considered.

1. Please add axes titles (both horizontal and vertical axes) to all the figures.

2. Table 1: Please remove the title of the column (Title 1).

3. Table 1: It would be more meaningful to add a column listing the n (%) of pharmacy schools teaching each of these concepts based on the authors' evaluation of the courses syllabi.

4. I think the discussion section starts from line 188 because the text from this line and in isn't describing the results, but rather discussing them and discussing pertinent literature. The results section should include solely the findings of this study without citations of previous studies or papers. Please move the subheading "Discussion" to before line 188.

5. From line 318, I suggest titling this section with something like "Proposed recommendations" to separate it from the discussion section after the limitations paragraph as typically the limitations paragraph should be the last part of a discussion section.

Author Response

This is a good study evaluating the extent and topics of the pharmacoeconomics courses in PharmD programs in the US. A few minor comments are suggested to be considered.

  1. Please add axes titles (both horizontal and vertical axes) to all the figures.

AUTHORS RESPONSE: Axes tiles were added to all the figures in the manuscript.

  1. Table 1: Please remove the title of the column (Title 1).

AUTHORS RESPONSE: The title of the Column (Title 1) was removed.

  1. Table 1: It would be more meaningful to add a column listing the n (%) of pharmacy schools teaching each of these concepts based on the authors' evaluation of the courses syllabi.

AUTHORS RESPONSE: We appreciate the reviewers' suggestions. However, assigning a specific percentage may pose challenges, as some schools may not explicitly mention certain topics in their syllabi, yet they could address them in some manner. For instance, even if a syllabus notes coverage of cost-effectiveness analysis, the discussion on Markov models might be incorporated within the broader context of CEA.

  1. I think the discussion section starts from line 188 because the text from this line and in isn't describing the results, but rather discussing them and discussing pertinent literature. The results section should include solely the findings of this study without citations of previous studies or papers. Please move the subheading "Discussion" to before line 188.

AUTHORS RESPONSE: We agree with the reviewer's comments and consequently started the discussion header before the start of Line 188. Now, the results section exclusively presents the findings of this study without referencing previous studies or papers.

  1. From line 318, I suggest titling this section with something like "Proposed recommendations" to separate it from the discussion section after the limitations paragraph as typically the limitations paragraph should be the last part of a discussion section.

AUTHORS RESPONSE: We express gratitude to the reviewers for their discerning eyes. To distinguish it from the discussion section, we have titled the portion starting from Line 318 as "Proposed Recommendations." In this section, we provide suggestions and guidance based on the study's findings, outlining specific actions, changes, or measures to address issues in pharmacoecoeconomics education and leverage opportunities identified in the research.

Reviewer 3 Report

Comments and Suggestions for Authors

The aim of the manuscript is to asses the extend to which accredited pharmacy colleges in the United States cover pharmacoeconomics contents within the didactic curriculum of their Doctor of Pharmacy programs. A systematic search of the curriculum websites of all 141 United States-based pharmacy programs holding accredited professional degree programs was carried to identify relevant pharmacoeconomics content.   

In general, the structure and concise working of the manuscript is adequate, which greatly facilitates its understanding by potential readers. On the other hand, the tables and the figures in the manuscript are adequate.

Comment: Have the authors considered including the names of the x and y axes of the graphs?

Author Response

The aim of the manuscript is to asses the extend to which accredited pharmacy colleges in the United States cover pharmacoeconomics contents within the didactic curriculum of their Doctor of Pharmacy programs. A systematic search of the curriculum websites of all 141 United States-based pharmacy programs holding accredited professional degree programs was carried to identify relevant pharmacoeconomics content.  

In general, the structure and concise working of the manuscript is adequate, which greatly facilitates its understanding by potential readers. On the other hand, the tables and the figures in the manuscript are adequate.

Comment: Have the authors considered including the names of the x and y axes of the graphs?

AUTHORS RESPONSE: We would like to thank the reviewers for their thoughtful comments and efforts towards improving our manuscript. In the revised manuscript, we added labels to both the x and y axes on each graph for clarity.

Reviewer 4 Report

Comments and Suggestions for Authors

Thank you for completing this research!

I am a bit concerned as to the necessity of this research given that pharmacoeconomics is a required component of the pharmacy curriculum. I also have concerns about the way in which you collected your data as information may not have been available or current on certain websites, meaning that these schools could have still been teaching pharmacoeconomics but it was "hidden" in other courses. 

Was there any attempt at contacting the 30 schools that did not have pharmacoeconomics content to confirm that they did not teach these concepts vs it simply being not included on their website?

I believe your research could be strengthened by perhaps comparing the pharmacoeconomic education of pharmacists vs other healthcare professionals, for example. 

Author Response

Thank you for completing this research!

I am a bit concerned as to the necessity of this research given that pharmacoeconomics is a required component of the pharmacy curriculum. I also have concerns about the way in which you collected your data as information may not have been available or current on certain websites, meaning that these schools could have still been teaching pharmacoeconomics but it was "hidden" in other courses.

 Was there any attempt at contacting the 30 schools that did not have pharmacoeconomics content to confirm that they did not teach these concepts vs it simply being not included on their website?

I believe your research could be strengthened by perhaps comparing the pharmacoeconomic education of pharmacists vs other healthcare professionals, for example.

AUTHORS RESPONSE: Thank you very much for taking the time to review this manuscript. Please find the detailed responses below and the corresponding revisions/corrections highlighted/in track changes in the re-submitted files.

Given the absence of research on the extent of pharmacoeconomic education in the United College and School of Pharmacy in almost a decade, coupled with changes in the delivery of pharmacoeconomics over the past ten years, our study is essential. One of its primary objectives is to provide updated insights into how pharmacoeconomics is currently integrated into curricula across accredited programs.

In response to the reviewer's comment on data collection, we have explained the challenges in the limitation section, acknowledging the encountered issues and outlining our strategies for overcoming them in future investigations. This is indicated in the statement below which was taken out of the manuscript.

“To the best of our knowledge, this is the only recent work that has investigated the state of pharmacoeconomics education in the United States regarding course objectives, number of credit hours, and syllabi content. However, our assessment does have some limitations worth acknowledging. First, several pharmacy programs did not publicly have their curriculum information posted on their main website, preventing us from assessing those programs. We did not contact the school representatives of such programs to obtain information on aspects of their curricula that were not accessible online. Secondly, our search methodology focused solely on identifying pharmacoeconomics-related content, meaning we may have missed other courses that cover concepts related to pharmacoeconomics but do not explicitly mention them. It is important to consider these limitations when interpreting the findings of our study. Future research could explore additional sources or methods to gather comprehensive data on pharmacoeconomics education in the United States, addressing these limitations and providing a more comprehensive understanding of the current state of pharmacoeconomics education. In future investigations, it is crucial to longitudinally assess student learning and retention within the realm of pharmacoeconomics.”

While we haven't considered comparing the pharmacoeconomic education of pharmacists with that of other healthcare professionals, we will certainly bear this aspect in mind in our future research endeavors.

Reviewer 5 Report

Comments and Suggestions for Authors

The work is very interesting. The data can provide very significant guidelines for further research. Methodologically, it is not sufficiently clearly explained how they were selected programs ( how they got all the programs). The paper does not mention the percentage of mandatory separate subjects. In the Fig. 2. Pharmacoeconomics course occurrence by professional year (n=107) explain what does P mean and indicate whether all programs last 3 years. They are not shown in the Figure 3, distribution for 5 and 6 credit hour. Does this mean that there were none?

The work would be richer if specific percentages were included, e.g:

-176 line although only about half of the reviewed programs' syllabi mentioned the application of pharmacoeconomics

-186 line skills were not a significant part of the descriptions

- 193 line Most pharmacoeconomics courses did not require a textbook

Too, term pedagogy replace with teaching.

Author Response

The work is very interesting. The data can provide very significant guidelines for further research. Methodologically, it is not sufficiently clearly explained how they were selected programs ( how they got all the programs). The paper does not mention the percentage of mandatory separate subjects. In the „Fig. 2. Pharmacoeconomics course occurrence by professional year (n=107)“ explain what does P mean and indicate whether all programs last 3 years. They are not shown in the Figure 3, distribution for 5 and 6 credit hour. Does this mean that there were none?

The work would be richer if specific percentages were included, e.g:

-176 line although only about half of the reviewed programs' syllabi mentioned the application of pharmacoeconomics

AUTHORS RESPONSE: Thank you very much for taking the time to review this manuscript. Please find the detailed responses below and the corresponding revisions/corrections highlighted/in track changes in the re-submitted files.

The n and % numbers were provided. The sentence now reads as follows “The majority of syllabi from the reviewed programs covered various methods of pharmacoeconomic analysis, although only about half (n=54; 49%) of the reviewed programs' syllabi mentioned the application of pharmacoeconomics.”

-186 line skills were not a significant part of the descriptions.

AUTHORS RESPONSE: Although the available syllabi did not explicitly discuss skills, we refrain from assigning a specific percentage, as it would be presumptuous on our part to do so solely based on our professional judgment.

- 193 line Most pharmacoeconomics courses did not require a textbook

AUTHORS RESPONSE: The n and % numbers were provided. The sentence now reads as follows. Most pharmacoeconomics courses (n=68; 61%) did not explicitly require a textbook. The courses that required one mostly adopted "Essential of Pharmacoeconomics"

Too, term pedagogy replace with teaching.

AUTHORS RESPONSE: The term pedagogy was replaced by teaching throughout the manuscript.

Round 2

Reviewer 4 Report

Comments and Suggestions for Authors

I appreciate the author's diligence in conducting this research